# Menopausal hormone therapy: Characterising users in an Australian national cross-sectional study

**Louiza S. Velentzis**[1,2]*, **Sam Egger**[1], **Emily Banks**[3], **Karen Canfell**[1,4]

**1** Daffodil Centre, The University of Sydney, a joint venture with Cancer Council NSW, Woolloomooloo, Sydney, New South Wales, Australia, **2** Melbourne School of Population and Global Health, Centre for Epidemiology and Biostatistics, University of Melbourne, Melbourne, Victoria, Australia, **3** National Centre for Epidemiology and Population Health, Research School of Population Health, Australian National University, Canberra, Australian Capital Territory, Australia, **4** Faculty of Medicine, Prince of Wales Clinical School, University of New South Wales, Sydney, New South Wales, Australia

* louizav@nswcc.org.au

**Data Availability Statement:** Data cannot be shared publicly due to restrictions imposed by the Cancer Council NSW Ethics Committee. Data can potentially compromise patient confidentiality. Data

## Abstract

Menopausal hormone therapy (MHT) is effective for menopausal symptoms, however, its use is also associated with risks of serious health conditions including breast, ovarian and endometrial cancer, stroke and venous thromboembolism. MHT-related health risks increase with longer durations of use. In Australia, while overall MHT use fell when risk-related findings were published in 2002, a significant number of women continue using MHT long-term. We aimed to examine socio-demographic, health-related and lifestyle characteristics in relation to post-2002 MHT use, and to compare use for <5 and ≥5 years. Data from 1,561 participants from an Australian, national, cross-sectional survey of women aged 50–69 in 2013 were analysed. Odds ratios (ORs) were calculated using logistic regression for characteristics related to overall MHT use post-2002 and multinomial logistic regression for associations between MHT duration of use [never/<5 years/≥5 years] and personal characteristics, adjusting for sociodemographic, reproductive, health and lifestyle factors. Post-2002 MHT use was associated with increasing age (p-trend<0.001), hysterectomy versus no hysterectomy (OR:2.55, 95%CI = 1.85–3.51), bilateral oophorectomy vs no oophorectomy (OR:1.66, 95%CI = 1.09–2.53), and ever- versus never-use of therapies other than MHT for menopausal symptoms (OR:1.93, 95%CI = 1.48–2.57). Women with prior breast cancer (OR:0.35, 95%CI = 0.17–0.74) and with more children (p-trend = 0.034) were less likely than other women to use MHT. Prior hysterectomy was more strongly associated with MHT use for ≥5 years than for <5 years (p = 0.004). Ever-use of non-MHT menopausal therapies was associated with MHT use for <5 years but not with longer-term use (p = 0.004). This study reinforces the need for MHT users and their clinicians to re-evaluate continued MHT use on an ongoing basis.

are available upon request from Dr. Louiza S Velentzis (e-mail address: louizav@nswcc.org.au) or from Mrs. Katie Armstrong, Cancer Research Program Specialist who can be contacted by email at: (Info@daffodilcentre.org) for researchers who meet the criteria for access to confidential data.

**Funding:** The study was funded by Cancer Council NSW https://www.cancercouncil.com.au/. The funder had no role in study design, data collection and analysis, decision to publish or preparation of the manuscript.

**Competing interests:** I have read the journal's policy and KC has the following competing interests: she is co-principal investigator of an investigator-initiated trial of cervical screening in Australia (Compass; ACTRN12613001207707 and NCT02328872), which is conducted and funded by the VCS Foundation (VCS), a government-funded health promotion charity. She is also an investigator of Compass New Zealand (ACTRN12614000714684), which was conducted and funded by Diagnostic Medlab (DML), now Auckland District Health Board. The VCS Foundation received equipment and a funding contribution from Roche Molecular Systems and Ventana USA and DML received equipment and a funding contribution for Compass from Roche Molecular Systems. However, neither KC nor her institution on her behalf (Cancer Council NSW) receives direct funding from industry for this trial or any other project. LSV, SE, and EB have no conflicts of interests to declare.

## Introduction

During the perimenopausal and postmenopausal periods, the decline in hormonal levels can lead to vasomotor symptoms in ≥55% of women [1]. Although the majority will experience mild to moderate symptoms [1], in a few women severe symptoms will have a significant impact on their quality of life. Menopausal hormone therapy (MHT), also referred to as hormone replacement therapy, is an effective first-line treatment for alleviating vasomotor symptoms in women who have been fully informed about the risks and benefits of use [2]. However, evidence from randomised trials and observational studies, particularly from 2002 onwards, demonstrated that use of MHT increases the risk of a range of conditions including breast, ovarian and endometrial cancer, stroke, and venous thromboembolism [3,4] and that these risks increase with increasing duration of use in postmenopausal women 50 years of age and older [5]. Taking this evidence into consideration, regulatory agencies changed their major recommendations around 2002–3, recommending use of MHT for menopausal symptoms only, in fully-informed women, using the use of the lowest dose of MHT possible for the shortest time [4,6]. The North American Menopause Society position statement in 2017 also stated that 'the risks differ for different women, depending on type, dose, duration of use, route of administration, timing of initiation, and whether a progestogen is needed'; it recommended that treatment should be individualized with periodic re-evaluation for the benefits and risks of MHT continuation [7].

Estimates from Australia in 2013–2014 have reported prevalence of current use of MHT in women in their fifties and sixties to be around 12–13% [8,9], after a significant decrease in use between 2001 and 2005 [10]. Using data from the Learning how Australians Deal with menopause sYmptoms (LADY) study, a cross-sectional survey of women conducted in 2013, we also previously reported that among current users, three-quarters of women had used MHT for 5 years or more. This is potentially concerning because many MHT-related health risks, particularly breast cancer, increase with the duration of MHT use [11]. Based on a recent meta-analysis of the worldwide evidence, the relative risk (RR) of breast cancer in current users (1–4 years) versus never-users of estrogen plus progestogen therapy (EPT) was 1.60, with excess risk nearly doubling to a RR of 2.08 for current use of 5–14 years. Although the excess risk associated with use of estrogen-only therapy (ET) was less, it was still significantly raised and also increased with duration [1–4 years: RR = 1.17; 5–14 years: RR = 1.33]. In Australia and much of the world, ET is usually prescribed to women who have had a hysterectomy whereas women with a uterus are prescribed a progestogen together with estrogen, to attenuate the estrogen-related risk of endometrial cancer.

Based on this evidence, characterising MHT users is important as their health risks and the need for clinical follow up vary according to their patterns of use. Moreover, evidence on who used MHT prior to 2002 may not reflect current users as guidance for use has changed over time and many women post-2002 may choose not to use MHT given the known risks. Using national Australian data from women aged 50–69 years of age in the LADY study, and taking into consideration the change in landscape since the publication of the first Women's Health Initiative Trial findings, the aim of this study was to a) characterise women starting MHT since 2002 in terms of socio-demographic, reproductive, health-related and lifestyle characteristics and b) investigate how MHT users of < 5 years differ from longer-term users of ≥ 5 years.

## Materials and methods

### Study participants

Data were obtained from the LADY study which has been described previously [8]. Briefly, during January and February 2013, women aged 50–69 years of age, residing in Australia were

sampled from the Medicare Australia enrolment database (Australia's national universal healthcare scheme) using an age-stratified, random sampling method. Women were invited by letter to complete a consent form and a questionnaire assessing the use of MHT and other menopausal therapies in addition to socio-demographic, health, and lifestyle characteristics. Questions on MHT use and duration were based on previously validated questions [12]. A total of 4,428 women participated in the study.

The LADY study received ethical approval from the Cancer Council NSW Human Research Ethics Committee on the 19th December 2011, project reference number 256. Participants completed and signed a consent form.

## Data and statistical analyses

A range of pre-specified characteristics of study participants were assessed for their relation to MHT use. Socio-demographic factors that were evaluated included: age; country of birth; highest educational qualification obtained; marital status; socio-economic indexes for Areas (SEIFA) [13]; Accessibility-Remoteness Index of Australia Plus (ARIA+) which is a measure of geographical remoteness [14]; income and occupation. Reproductive and health-related characteristics assessed included: age at menarche; use of hormonal contraceptives; number of births; length of time breastfeeding; ever having a hysterectomy; ever having a bilateral oophorectormy; family history of breast or ovarian cancer; history of breast cancer; history of ovarian cancer and mammography screener frequency (based on responses to questions on mammography: "Have you ever been for a breast screening mammogram", "If yes, about how many years ago was your last breast screening mammogram", and "How many times have you been for a breast screening mammogram in the past 10 years?" this variable was categorised as regular/over screener vs under-/never-screener). Lifestyle behaviours that were assessed included: body mass index (BMI); alcohol intake (drinks per week); smoking; exercise level [nil to high activity as calculated by Nunez et al [15]; and use of therapies other than MHT for menopausal symptoms (e.g. evening primrose oil, multivitamins, vitamin E, exercise, acupuncture, Chinese herbs, change in diet, other).

To investigate which factors were associated with the use of MHT during a time period where the health risks of MHT were widely known (i.e. after 2002) we excluded all women who were most likely to use MHT before 2002. This included women who were over 50 years of age in 2002 as well as women aged 50 or younger who reported MHT use at that time (n = 2,393). Logistic regression was used to assess the association between MHT use after 2002 and participant characteristics, adjusting for all the socio-demographic, reproductive and health-related characteristics listed in the above paragraph (and also in Tables 1–3). The associations between MHT use [(never-use, use for <5 years, and use for ≥5 years)] and the same characteristics (listed above and also in S1–S3 Tables) were then examined using multinomial logistic regression, with never users set as the base-outcome category. For the multinomial regression, current use of MHT for <5 years could not be determined based on duration of use because of the cross-sectional nature of the survey, i.e. a proportion of these current users at the time the study questionnaire was completed are likely to have continued their MHT use in the future and become longer-term users. We therefore defined this category based on women's reported use of MHT for <5 years between 2002 until 2013. A complete case approach was adopted for the analysis, thus participants with missing/unknown information on MHT use or any of the factors examined were excluded from analyses (n = 286) (although 'prefer not answer' responses were treated as a valid category where that response option was available on the questionnaire). Estimates of effect were measured by adjusted odds ratios (ORs) and 95% confidence intervals (CIs) from the regression models. For the multinomial

**Table 1. Associations between socio-demographic characteristics and menopausal hormone therapy use between users initiating MHT after 2002 and never users (younger than 50 in 2002) among Australian women aged 50–69 years old.**

| Characteristic | Never users N = 1214 | Post-2002 users N = 347 | Adjusted OR (95%CI)^ | *p-global; p-trend* |
|---|---|---|---|---|
| **Age group (years)** | | | | |
| 50–54 | 692 (82) | 148 (18) | Ref. | |
| 55–59 | 472 (74) | 168 (26) | 1.72 (1.31–2.26) | <0.001 |
| 60–64 | 50 (62) | 31 (38) | 3.03 (1.77–5.19) | <0.001 |
| **Country of Birth** | | | | |
| Australia | 917 (79) | 249 (21) | Ref. | |
| English-speaking country | 164 (73) | 61 (27) | 1.39 (0.97–1.99) | 0.201 |
| Non-English speaking country | 133 (78) | 37 (22) | 1.09 (0.69–1.71) | |
| **Highest qualification** | | | | |
| No school | 64 (68) | 30 (32) | Ref. | |
| High school/trade/apprenticeship | 425 (21) | 114 (21) | 0.52 (0.30–0.91) | 0.136 |
| Certificate diploma | 330 (78) | 92 (22) | 0.54 (0.30–0.96) | |
| University degree or higher | 395 (78) | 111 (22) | 0.52 (0.29–0.94) | |
| **Marital status** | | | | |
| Single-never married | 93 (82) | 21 (18) | Ref. | 0.571 |
| Married/partnered | 932 (77) | 279 (23) | 1.26 (0.71–2.23) | |
| Single-divorced, separated, widowed | 189 (80) | 47 (20) | 1.07 (0.57–2.03) | |
| **Socioeconomic Index for Areas quintile** | | | | |
| 1 - lowest SES | 154 (78) | 43 (22) | Ref. | |
| 2 | 210 (80) | 53 (20) | 0.81 (0.50–1.33) | 0.857 |
| 3 | 228 (76) | 73 (24) | 1.02 (0.63–1.65) | |
| 4 | 242 (78) | 70 (22) | 0.92 (0.56–1.49) | |
| 5 - highest SES | 380 (78) | 108 (22) | 0.98 (0.60–1.59) | |
| **ARIA plus** | | | | |
| Major City | 742 (78) | 209 (22) | Ref. | |
| Inner Regional | 293 (76) | 91 (24) | 1.16 (0.84–1.62) | 0.643 |
| Outer regional/remote/very remote | 179 (79) | 47 (21) | 1.13 (0.74–1.73) | |
| **Income (per year)** | | | | |
| ≤ $25,000 | 317 (78) | 88 (22) | Ref. | |
| $25,001-$100,000 | 337 (78) | 98 (23) | 1.13 (0.79–1.64) | 0.078 |
| ≥ $100,001 | 290 (75) | 99 (26) | 1.39 (0.92–2.08) | |
| Prefer not to answer | 270 (81) | 62 (19) | 0.82 (0.55–1.22) | |
| **Occupation** | | | | |
| Employed | 930 (78) | 259 (22) | Ref. | |
| Unemployed | 35 (76) | 11 (24) | 1.13 (0.54–2.39) | 0.927 |
| Not in labour force (e.g. retired) | 249 (76) | 77 (24) | 1.04 (0.75–1.47) | |

^ Adjusted ORs obtained from logistic regression model.

The dependent variable in the logistic regression model was MHT use after 2002 (yes vs no) while the independent variables were the characteristics listed in Tables 1–3.

model, separate global p-values were estimated for the association of each characteristic with MHT use for <5 years and ≥5 years and to assess whether ORs differed between the two groups of users. A value of <0.05 was considered statistically significant. Stata 11.0 software (StataCorp) was used to conduct all statistical analysis.

**Table 2. Associations of reproductive/health-related characteristics and menopausal hormone therapy use between users initiating MHT after 2002 and never users (younger than 50 in 2002) among Australian women aged 50–69 years old.**

| Characteristic | Never users N = 1214 | Post-2002 users N = 347 | Adjusted OR (95%CI)^ | p-global; p-trend |
|---|---|---|---|---|
| **Age at menarche (years)** | | | | |
| 12–14 | 815 (77) | 244 (23) | Ref. | |
| ≤ 11 | 189 (78) | 52 (22) | 0.88 (0.61–1.28) | 0.339 |
| ≥ 15 | 210 (81) | 51 (20) | 0.77 (0.53–1.01) | 0.528 |
| **Use of hormonal contraceptives** | | | | |
| Never | 113 (80) | 29 (20) | Ref. | 0.129 |
| Ever | 1101 (78) | 318 (22) | 1.12 (0.68–1.85) | |
| **Number of births** | | | | |
| None | 164 (78) | 47 (22) | Ref. | |
| 1–2 | 597 (75) | 197 (25) | 0.75 (0.43–1.40) | 0.032 |
| ≥ 3 | 453 (82) | 103 (19) | 0.54 (0.29–1.01) | 0.034 |
| **Length of time breastfeeding (months)** | | | | |
| None | 281 (79) | 77 (22) | Ref. | |
| ≤ 11 | 298 (73) | 113 (28) | 1.58 (0.96–2.61) | 0.092 |
| ≥ 12 | 635 (80) | 157 (20) | 1.18 (0.72–1.94) | 0.103 |
| **Hysterectomy** | | | | |
| No | 1030 (82) | 232 (18) | Ref. | |
| Yes | 184 (62) | 115 (39) | 2.55 (1.85–3.51) | <0.001 |
| **Bilateral oophorectomy** | | | | |
| No | 1187 (80) | 305 (20) | Ref. | |
| Two ovaries removed | 27 (39) | 42 (61) | 1.66 (1.09–2.53) | 0.002 |
| **Family history of breast/ovarian cancer** | | | | |
| No | 791 (78) | 230 (23) | Ref. | |
| Yes | 423 (78) | 117 (22) | 0.90 (0.68–1.18) | 0.318 |
| **History of breast cancer** | | | | |
| No | 1150 (77) | 338 (23) | Ref. | |
| Yes | 64 (88) | 9 (12) | 0.35 (0.17–0.74) | 0.001 |
| **History of ovarian cancer** | | | | |
| No | 1204 (78) | 343 (22) | Ref. | |
| Yes | 10 (71) | 4 (29) | 1.09 (0.30–4.03) | 0.100 |
| **Mammography screener frequency** | | | | |
| Regular/ over-screens | 230 (82) | 52 (18) | Ref. | |
| Under-screens/never screens | 984 (80) | 295 (23) | 1.30 (0.91–1.85) | 0.153 |

^ Adjusted ORs obtained from logistic regression model.

The dependent variable in the logistic regression model was MHT use after 2002 (yes vs no) while the independent variables were the characteristics listed in Tables 1–3.

## Results

From 4428 LADY study participants after exclusions (Fig 1) complete data for all factors analysed were available for 1,561 women of whom 1,214 (78%) were never-users, and 347 (22%) were users from 2002 onwards. The average age of participants was 55 years (median 55); non-MHT users were 55 (median 54 years) and MHT users were 56 years old (median 56).

Tables 1–3 present the adjusted associations between use of MHT after 2002 and socio-demographic, reproductive and health-related characteristics, and lifestyle behaviours. Post-2002 use of MHT increased with increasing age (p-trend<0.001). Parity was inversely

**Table 3. Associations of lifestyle behaviours and menopausal hormone therapy use between users initiating MHT after 2002 and never users (younger than 50 in 2002) among Australian women aged 50–69 years old.**

| Characteristic | Never users N = 1214 | Post-2002 users N = 347 | Adjusted OR (95%CI)^ | p-global; p-trend |
|---|---|---|---|---|
| **Body mass index (kg/m²)** | | | | |
| ≤24.9 (underwt/normal) | 513 (77) | 157 (23) | Ref. | 0.268 |
| 25–29.9 (overweight) | 393 (79) | 107 (21) | 0.81 (0.60–1.09) | 0.342 |
| ≥ 30.0 (obese) | 308 (89) | 83 (21) | 0.79 (0.57–1.12) | |
| **Alcohol (drinks per week)** | | | | |
| 0 | 237 (78) | 66 (22) | 0.90 (0.62–1.31) | 0.164 |
| 1–3 | 473 (81) | 111 (19) | Ref. | |
| ≥ 4 | 504 (75) | 170 (25) | 1.21 (0.85–1.73) | 0.129 |
| **Smoking status** | | | | |
| Never | 725 (78) | 206 (22) | Ref. | |
| Ex-smoker | 390 (78) | 112 (22) | 0.95 (0.71–1.27) | 0.933 |
| Current | 99 (77) | 29 (23) | 1.00 (0.60–1.66) | |
| **Exercise level** | | | | |
| Nil activity | 317 (79) | 86 (21) | Ref. | |
| Low activity | 470 (76) | 149 (24) | 1.08 (0.77–1.50) | 0.686 |
| Moderate activity | 361 (79) | 98 (21) | 0.90 (0.63–1.30) | |
| High activity | 66 (83) | 14 (18) | 0.84 (0.43–1.65) | |
| **Ever use of therapies other than MHT for menopausal symptoms** | | | | |
| No | 695 (84) | 134 (16) | Ref. | <0.001 |
| Yes | 519 (71) | 213 (29) | 1.93 (1.48–2.57) | |

^ Adjusted ORs obtained from logistic regression model.

The dependent variable in the logistic regression model was MHT use after 2002 (yes vs no) while the independent variables were the characteristics listed in Tables 1–3.

associated with MHT use (p-trend = 0.034). Having a hysterectomy or a bilateral oophorectomy were positively associated with MHT use [OR:2.55 (95%CI:1.85–3.51); p<0.001 and OR:1.66 (95%CI 1.09–2.53); p = 0.002, respectively] as well as ever use of therapies other than MHT for menopausal symptoms (OR:1.93, 95%CI 1.48–2.57; p<0.001). Also, women with a prior history of breast cancer were less likely to use MHT compared to women with no history (OR:0.35, 95%CI = 0.17–0.74, p = 0.001).

When taking into consideration the duration of MHT use, compared to women with a uterus, women with a hysterectomy had twice the odds of using MHT for < 5years (OR:2.04; 95%CI 1.42–2.93), but over 5 times the odds of using MHT for 5 years or more (OR:5.07; 95% CI 2.87–8.94); these ORs differed significantly [p = 0.004] (S2 Table). Compared to women who had never used alternative therapies, ever-users had 2.28 times the odds of using MHT for < 5 years (95%CI = 1.70–3.06), but similar odds of using MHT for 5 years or more (OR:1.00; 95%CI 0.61–1.66) [p = 0.004 for difference between ORs] (S3 Table).

## Discussion

To our knowledge this is the first study to investigate specific determinants of MHT use evaluated wholly after the release of the WHI and other results, from 2002 onwards, with participants drawn from a nationally-representative cohort. In this Australian cross-sectional study of women aged 50–69 years, increasing age, having a prior hysterectomy or bilateral oophorectomy, and ever use of therapies for menopausal symptoms other than MHT were associated

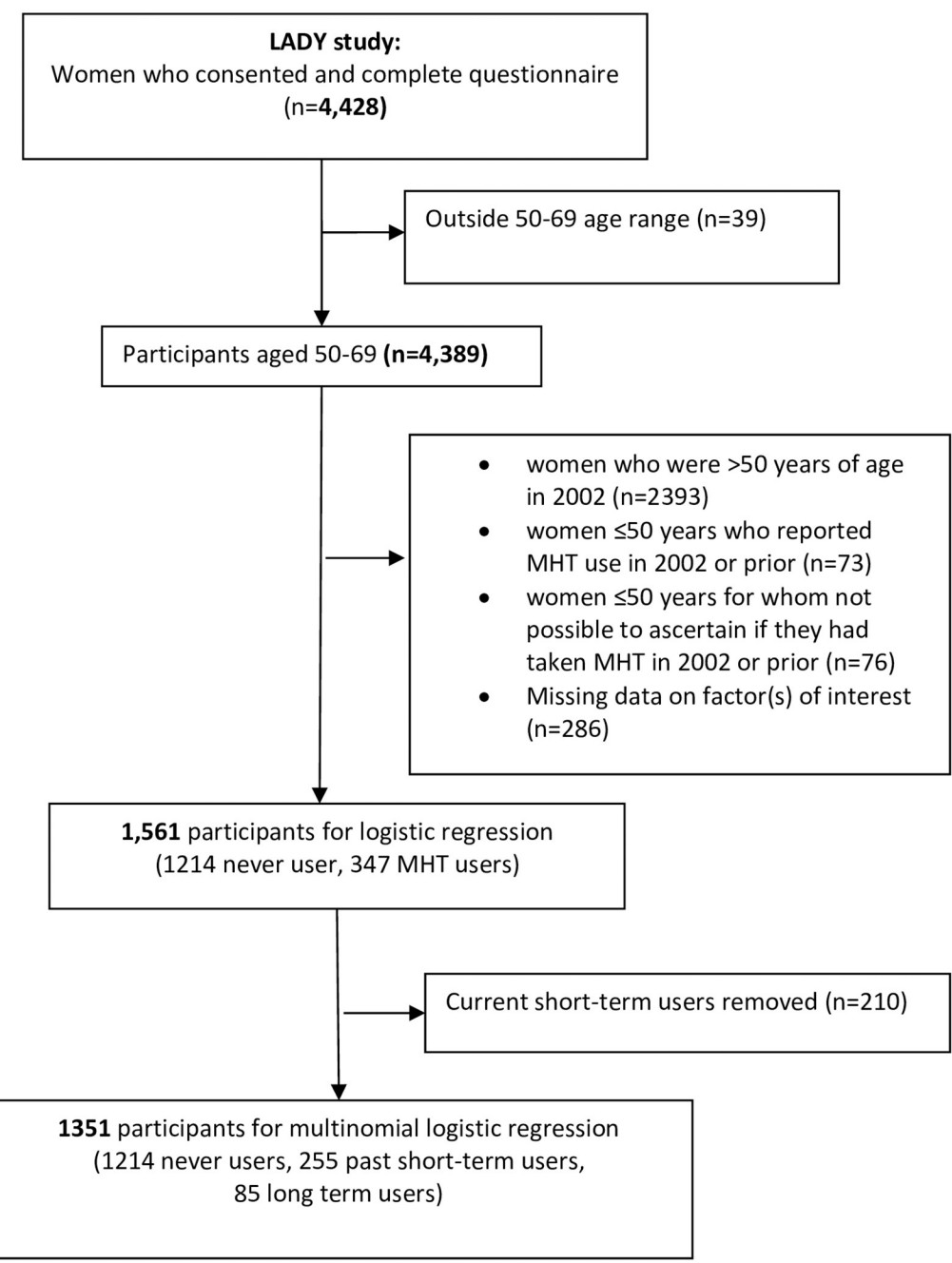

**Fig 1. Flow diagram showing participation in the current analyses.**

with MHT use commencing after 2002. Women who had given birth to more children and those reporting a history of breast cancer, were less likely to be MHT users. Prior hysterectomy was more strongly associated with MHT use for ≥5 years than shorter term use, and ever use of menopausal therapies other than MHT was associated with MHT use for <5 years but not longer-term use.

As expected, MHT use was positively associated with a prior bilateral oophorectomy or hysterectomy. Removal of the ovaries leading to surgical menopause can cause the sudden onset of menopausal symptoms, which in some cases is severe. Surgical menopause has been consistently associated with MHT use in a number of studies [16–20]. We also found that women who had had a hysterectomy were considerably more likely to use MHT for 5 years or longer than those with an intact uterus. Long-term MHT users who have had a hysterectomy would most likely be using ET; in an earlier analysis of the LADY cohort we found that 77% of hysterectomised current MHT users reported ET use [8]. Although the risk of breast cancer associated with ET is less compared to the risk associated with EPT use, long-term users should still be advised that there is a higher risk compared to no use.

Within this group of women 50+ years, age was positively associated with MHT and use was found to increase with increasing age. Initiation of MHT use in women of different age groups could be complex and multifactorial–underlying considerations include the possibility of hysterectomy, the emergence of other health concerns or co-morbidities as women age, patient preferences and/or clinician recommendations. Based on data from the current study the reasons behind this trend therefore cannot be directly ascertained. A previous survey investigating influences contributing to the initiation of MHT in women aged 60 and older conducted in the US reported that compared to women who did not initiate MHT, those who did were more likely to have had a hysterectomy at $\geq$ 60 years of age, report good health, exercise more and have beliefs about the benefits of MHT use in relation to reduced risk of bone fractures and heart disease but less likely to believe that MHT use increased the risk of breast cancer [21]. Additionally, in our cohort we expect that a proportion of women aged 50–54 and a smaller proportion aged 55–59 year, will be pre-menopausal and therefore not at the stage that they would consider the use of MHT. In parallel, older women are not only more likely to be post-menopausal but would also have a longer time during which to initiate MHT and therefore a higher chance of becoming an ever user of MHT. Both these factors together are probably driving the observed relationship of increased use of MHT with increasing age. Overall, it should be noted that we adjusted for age in our analyses, so the difference we observe in the other characteristics shouldn't be confounded by age differences in MHT use.

We also found that increasing parity was associated with reduced odds of MHT use. Although the reasons behind this result are unclear, it is possible that use of MHT is more likely by women who experience premature menopause (before age 40) or early menopause (40–44 years of age) and this risk has been found to be higher in nulliparous women compared to women with 2 or more children according to a pooled analysis of 9 international studies [22]. Furthermore, in this cohort women with a history of breast cancer had reduced odds of MHT use. There are only a few studies that have investigated the impact of MHT use by women previously diagnosed with breast cancer. Results from two randomised non-placebo-controlled trials of systemic MHT in women with a breast cancer history were both stopped prematurely [23,24]. In the Stockholm trial, after 10.8 years of follow-up, and a mean duration of 2.6 +/- 1.2 years of MHT use by the intervention group, there was a significant increase in contra-lateral breast cancer in MHT users compared to the placebo group [HR = 3.6; (95% CI 1.2–10.9) [23]. In the HABITS trial, after a median follow-up of 4 years and a median duration of 2 years of MHT by the intervention group, there was a significantly elevated risk of a first breast cancer event in the MHT arm compared to the no-MHT arm [HR = 2.4 (95% CI = 1.3 to 4.2)] [24]. In LIBERATE, a double-blind randomised controlled trial of tibolone versus placebo, after a median follow-up of 3.1 years and a median duration of tibolone for 2.7 years, women in the intervention arm had a 40% increased risk of a breast cancer recurrence, compared with the placebo group [HR 1·40, 95% CI 1·14–1·70; p = 0·001] [25]. Taken together, the

evidence therefore does not support the use of systemic MHT for menopausal symptoms by women with a history of breast cancer. In terms of topical vaginal MHT use, there are no clinical trials investigating the risk of recurrence associated with their use. However, a collaborative analysis of 58 studies found that the use of vaginal estrogens was not associated with an increased risk of breast cancer, regardless of duration of use [11].

MHT has been shown to be an effective treatment for menopausal symptoms, however, many women seek alternatives; they may prefer nonhormonal therapies because they are concerned of the risks, want to use preparations that they perceive as more natural, may want to stop MHT or they have a pre-existing condition which is contraindicated for MHT use (e.g. history of breast cancer or endometrial cancer, active liver disease, a prior thromboembolism, stroke, coronary heart disease, dementia, hypertriglyceridemia, etc [7]. In this study, 39% of all participants reported use of therapies other than MHT for menopausal symptoms. Compared to never users women who used therapies other than MHT had higher odds of using MHT for less than 5 years. Of the available alternative nonhormonal treatments only a few are evidence-based while there are some complementary and alternative medicines (CAM) which have uncertain safety profiles and can interact adversely with certain medications [26,27]. Although a clinician can provide the required guidance, studies have shown that the proportion of women who report obtaining their information about the menopause from the media (internet, TV, magazines, newspapers) and friends, family members or social contacts is equal to or higher than the proportion of women who refer to their healthcare provider [28,29].

MHT use was self-reported in this study. Previous work has shown a very good agreement between data from prescription records and questionnaire data on MHT, using the same questions as in the LADY questionnaire (based on the Million Women Study) [12]. The sample size of MHT users was also relatively small in some subgroups according to the characteristics assessed; we note, for example that only 9 MHT users had a history of breast cancer although analysis of the relation with this characteristic still yielded a significant finding overall (P = 0.001). Despite these limitations, the study has a number of strengths. It draws on a population-based sample of women who were randomly selected from all Australian states, recruitment of participants was conducted from a single source—the Medicare Australia enrolment database, and a wide range of demographic, health and lifestyle characteristics were available for analysis.

Although a number of studies have investigated determinants of MHT use, very few were conducted entirely after the release of the initial findings from the WHI and other influential studies, from 2002 onwards [16,17]. Overall, previous studies have reported associations between current MHT use with higher education or more years of schooling [18,19,30,31]; ever use of hormonal contraceptives [18,19,31], alcohol consumption [16,31], and smoking [16,31]. In our study, these characteristics were not found to be significantly related to MHT use. This could be due to differences in the outcome variable used in the analyses and/or the size of the cohorts, the populations studied, differences in the time period the cohort was questioned, and other factors.

In the most recent Cochrane systematic review on the effects of MHT use in peri/postmenopausal women based on 22 randomised controlled trials, in addition to health risks from use of MHT for over 5 years (e.g. breast cancer), even short-term use of MHT was associated with health risks [5]. The authors reported an increase in the risk of stroke after 3 years of combined MHT use, an increase in the risk of a cardiovascular outcomes after 1–2 years of MHT, and in women over 65 years old an increase in the risk of dementia after 4 years of combined MHT use [5].

## Conclusions

Findings from this study combined with evidence from the literature on the health risks of MHT highlight the need for MHT users and their clinicians to keep re-evaluating whether continued MHT use is required by weighing up the risks and benefits of use. In addition to MHT, a number of CAMs are available and commonly used by menopausal women. As some of these can be of concern, guidance needs to be provided towards evidence-based alternative therapies for alleviating menopausal symptoms, if MHT cannot be used.

## Supporting information

**S1 Table. Associations between socio-demographic characteristics and menopausal hormone therapy use after 2002, between short-term users† (<5 years) and long-term users (≥5 years)‡ compared to never users, among Australian women aged 50–69 years old.** (DOCX)

**S2 Table. Associations of reproductive/health-related characteristics and menopausal hormone therapy use after 2002, between short-term users† (<5 years) and long-term users (≥5 years)‡ compared to never users, among Australian women aged 50–69 years old.** (DOCX)

**S3 Table. Associations of lifestyle behaviours and menopausal hormone therapy use after 2002, between short-term users† (<5 years) and long-term users (≥5 years)‡ compared to never users, among Australian women aged 50–69 years old.** (DOCX)

## Acknowledgments

We would like to thank all women who participated in the LADY study; their contribution is much appreciated.

## Author Contributions

**Conceptualization:** Louiza S. Velentzis, Karen Canfell.

**Data curation:** Louiza S. Velentzis.

**Formal analysis:** Louiza S. Velentzis.

**Funding acquisition:** Louiza S. Velentzis, Karen Canfell.

**Investigation:** Louiza S. Velentzis.

**Methodology:** Louiza S. Velentzis, Sam Egger, Emily Banks, Karen Canfell.

**Supervision:** Sam Egger, Karen Canfell.

**Writing – original draft:** Louiza S. Velentzis.

**Writing – review & editing:** Louiza S. Velentzis, Sam Egger, Emily Banks, Karen Canfell.

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
