## [Decision Letter · Decision Letter 0]

11 Jun 2021

Menopausal hormone therapy: characterising users in an Australian national cross-sectional study

PONE-D-21-15078

Dear Dr. Velentzis,

We’re pleased to inform you that your manuscript has been judged scientifically suitable for publication and will be formally accepted for publication once it meets all outstanding technical requirements.

Kind regards,

Martha Asuncion Sánchez-Rodríguez, PhD

Academic Editor

PLOS ONE

Additional Editor Comments (optional):

The research is interesting and relevant, I suggest attending the observations of reviewer 1 and consider the suggestion of reviewer 2 to improve the manuscript.

Reviewers' comments:

Reviewer's Responses to Questions

**Comments to the Author**

1. Is the manuscript technically sound, and do the data support the conclusions?

Reviewer #1: Yes

Reviewer #2: Yes

2. Has the statistical analysis been performed appropriately and rigorously? 

Reviewer #1: Yes

Reviewer #2: Yes

3. Have the authors made all data underlying the findings in their manuscript fully available?

Reviewer #1: No

Reviewer #2: No

4. Is the manuscript presented in an intelligible fashion and written in standard English?

Reviewer #1: Yes

Reviewer #2: Yes

5. Review Comments to the Author

Reviewer #1: Important study with participants from a nationally representative cohort. Well-structured summary, presenting context, method, main results and conclusion. I suggest checking the sum of the percentages in line 145 (78% + 23% = 101%). Relevant discussion, comparing the findings with international studies. The authors pointed out the limitations of the study. The conclusion is consistent with the study carried out.

Reviewer #2: The article is of potential interest and the findings were clear-cut. I only have a few specific concerns.

In results section I suggest to include the age at which menopause occurred and in discussion section the association between age and MHT use should be discussed.

6. PLOS authors have the option to publish the peer review history of their article (what does this mean?). If published, this will include your full peer review and any attached files.

Reviewer #1: **Yes: **Fernanda Piana Santos Lima de Oliveira

Reviewer #2: **Yes: **Renata Saucedo

---

## [Editor Report · Acceptance letter]

2 Aug 2021

PONE-D-21-15078 

Menopausal hormone therapy: characterising users in an Australian national cross-sectional study 

Dear Dr. Velentzis:

I'm pleased to inform you that your manuscript has been deemed suitable for publication in PLOS ONE. Congratulations! Your manuscript is now with our production department. 

Kind regards, 

on behalf of

Dr. Martha Asuncion Sánchez-Rodríguez 

Academic Editor

PLOS ONE